# Resistance to *Frankliniella occidentalis* during Different Plant Life Stages and under Different Environmental Conditions in the Ornamental Gladiolus

**DOI:** 10.3390/plants13050687

**Published:** 2024-02-29

**Authors:** Dinar S. C. Wahyuni, Peter G. L. Klinkhamer, Young Hae Choi, Kirsten A. Leiss

**Affiliations:** 1Plant Science and Natural Products, Institute of Biology (IBL), Leiden University, Sylviusweg 72, 2333BE Leiden, The Netherlands; dinarsari_cw@staff.uns.ac.id (D.S.C.W.); p.g.l.klinkhamer@biology.leidenuniv.nl (P.G.L.K.); 2Pharmacy Department, Faculty Mathematics and Natural Sciences, Universitas Sebelas Maret, Jl. Ir. Sutami 36A, Surakarta 57126, Indonesia; 3Natural Products Laboratory, Institute of Biology, Leiden University, Sylviusweg 72, 2333BE Leiden, The Netherlands; y.choi@chem.leidenuniv.nl; 4College of Pharmacy, Kyung Hee University, Seoul 02447, Republic of Korea; 5Business Unit Horticulture, Wageningen University and Research Center, Postbus 20, 2665ZG Bleiswijk, The Netherlands

**Keywords:** plant development stages, *Frankliniella occidentalis*, climate- and field-grown plants, Gladiolus, eco-metabolomics

## Abstract

The defense mechanisms of plants evolve as they develop. Previous research has identified chemical defenses against Western flower thrips (WFT) in *Gladiolus* (*Gladiolus hybridus* L.). Consequently, our study aimed to explore the consistency of these defense variations against WFT across the various developmental stages of *Gladiolus* grown under different conditions. Thrips bioassays were conducted on whole plants at three developmental stages, using the Charming Beauty and Robinetta varieties as examples of susceptible and resistant varieties, respectively. Metabolomic profiles of the leaves, buds and flowers before thrips infestation were analyzed. The thrips damage in Charming Beauty was more than 500-fold higher than the damage in Robinetta at all plant development stages. Relative concentrations of triterpenoid saponins and amino acids that were associated with resistance were higher in Robinetta at all plant stages. In Charming Beauty, the leaves exhibited greater damage compared to buds and flowers. The relative concentrations of alanine, valine and threonine were higher in buds and flowers than in leaves. The Metabolomic profiles of the leaves did not change significantly during plant development. In addition, we cultivated plants under different environmental conditions, ensuring consistency in the performance of the two varieties across different growing conditions. In conclusion, the chemical thrips resistance markers, based on the analysis of vegetative plants grown in climate rooms, were consistent over the plant’s lifetime and for plants grown under field conditions.

## 1. Introduction

Plant defenses are not fixed throughout a plant’s life. Major changes occur depending on growing conditions, plant development and the level of biotic and abiotic stress. For breeders, such changes may present a problem when they want to detect robust chemical markers for resistance in their breeding programs.

Plant resistance to herbivores has mostly been studied under controlled conditions in growth cabinets or climate rooms to minimize the effects of external variables on the plant metabolome. Under laboratory conditions, photoperiod, light intensity, temperature and humidity are controlled, whereas in the field, those conditions are highly variable. These external variables may thus cause variation in the levels of defense compounds and consequently affect plants’ resistance to herbivores. For instance, the concentration of triterpenoid saponins in plants is affected by habitat, season, plant age, light, temperature and water [1]. Amino acid levels were reported to depend on light conditions [2]. Also, drought affects amino acid contents and, through this, herbivore feeding performance [3].

During a plant’s lifetime, major changes in its defense system occur. This can be the result of aging tissues [4], or these changes can be associated with developmental switches such as from seedling to vegetative stage or from vegetative to flowering stage [5]. Generally, it is assumed that plant parts that most strongly contribute to s fitness are defended best [6]. For instance, young leaves are, in general, better protected from generalist herbivores than older leaves [7], and buds and flowers are better protected than leaves [8]. The ultimate choice of herbivores will be determined by both the nutritional value of the tissue and its level of defense. While for herbivores, such as thrips, young flowers with pollen can have high nutritional value [9], they may at the same time be better protected and have accumulated higher defense levels than other plant tissues such as leaves [9]. The effect of developmental stage or plant age on resistance has been well studied for a number of insect herbivores, including Western flower thrips (*Frankliniella occidentalis*, WFT). The WFT preference pattern is not fully consistent across plant species. In a greenhouse study with *Impatiens walleriana*, the rank order of WFT preference was 1. plants with flower buds, 2. plants with fully opened flowers with pollen, 3. plants with fully opened flowers without pollen and 4. plants with foliage without flowers [10]. In *Calystegia sepium,* WFT numbers increased during bud development and opening and reached a peak just before flowers started to wilt [11]. In both *Impatiens wallerana* and *Calystegia sepium,* WFT preferred flowers over leaves [10,11]. In tomato [12] and in Senecio [13], WFT damage was greater in older leaves. In tomato, this difference became stronger after external application of JA. Although from an evolutionary point of view, it makes sense that tissues that contribute less to fitness are not optimally defended, this presents a problem to growers. While high infestation levels on older leaves may not reduce flower or seed production, they may lead to unmarketable products or higher levels of virus infections as, e.g., in the case of thrips [14].

For plant breeders, potential changes in the plant’s defense system during plant development presents a problem because selection in breeding programs is based on the analyses of early life stages. The question is whether or not predictions of resistance in young plants are good indicators of resistance later in life. Especially, for herbivores that show a clear preference for particular plant organs such as buds, flowers or seeds this question is highly relevant. In this paper we studied the defenses of *Gladiolus* against WFT at three developmental stages and under different growing conditions. WFT is one of the most serious pests of agricultural and horticultural crops worldwide [15], causing losses of millions of euros. WFT is highly polyphagous, invading fruit, vegetables and ornamentals [16]. Thrips have piercing–sucking mouthparts which allow them to feed on different types of plant cells [17]. After sucking up the cell’s content, these fill with air, leading to the characteristic silver damage. Moreover, they are vectors of viral diseases [14].

In *Gladiolus* too, thrips infestation presents a severe problem. Plant breeders are in need of morphological or chemical markers to assist their breeding programs and to make full use of the natural variation that is present in *Gladiolus* with respect to thrips resistance. In earlier work, we investigated the differences in WFT resistance of various *Gladiolus* varieties [18] grown under climate room conditions. We detected, in a multivariate analysis of NMR data, signals related to thrips resistance. These were a signal at δ 0.90 ppm linked to triterpenoid saponins and the amino acids alanine and threonine. Subsequent correlation analyses gave significant relationships with the signal of 0.90 ppm, linked to triterpenoid saponins, alanine and threonine. All these signals were highly correlated among each other and with density of papillae [18]. Most likely these defence compounds are produced and/or stored in the extra cuticular papillae.

The objective of this study was to investigate the effect of plant developmental stages and environmental conditions on plant resistance to WFT. For this we investigated *Gladiolus* plants grown under natural field conditions of a plant breeder, plants transferred from the field (to a climate room and plants grown during the whole experiment in a climate room). The vegetative life stage comprises about 80% of the total life cycle of Gladiolus. However, success in later developmental stages of the plants is crucial for bulb and flower production. We, therefore, compared metabolomic profiles and WFT infestation for three developmental stages: vegetative, generative stage with buds and generative stage with flowers.

For our experiments we used the *Gladiolus* varieties Robinetta and Charming Beauty which in the vegetative stage were shown to be highly resistant and susceptible to WFT, respectively [18]. We specifically addressed the following questions: Do Robinetta and Charming Beauty show consistent differences in WFT resistance over all development stages?Does WFT damage differ between plant organs?Does WFT damage to leaves differ among plant development stages?Are differences between the metabolomic profiles of Robinetta and Charming Beauty consistent across developmental stages?Do the concentrations of defence compounds related to WFT resistance differ among plant organs?Do the concentrations of compounds that were related to WFT resistance alter with the development stages of the plant?Are the metabolic profiles of the plants dependent on the environmental conditions?And if so: Is there a change in the concentration of compounds related to thrips resistance?

## 2. Results

### 2.1. Thrips Whole Plant Bioassay

**Differences in total WFT damage between the two varieties**. While Charming Beauty showed considerable damage on flowers and leaves (Figure 1), hardly any damage occurred in Robinetta at all developmental stages. Total WFT damage differed significantly between Charming Beauty and Robinetta (U = 55.000, df = 1, *p* = 0.000) (Figure 2). The average total damage across all three developmental stages was: 565.22 ± 77.4 mm^2^ in Charming Beauty and 3.3 ± 1.9 mm^2^ in Robinetta.

**WFT Damage in Different Plant Organs.** In Charming Beauty damage to buds accounted for 35% of the total damage in the bud stage. Damage to flowers accounted for 16% from the total damage in this stage, while no damage to buds occurred in this stage. In Robinetta damage to all plant organs was low and in buds and flowers it was even zero (Figure 2).

**WFT Damage on Leaves at Different Plant-Stages**. WFT damage on leaves differed significantly among the three plant development stages in Charming Beauty (F = 16.593, df = 2, *p* = 0.023) (Figure 2). Damage in the vegetative stage was two times higher than in the generative stage with buds or flowers. In Robinetta WFT damage at all three developmental stages were close to zero and did not differ significantly developmental stages (H = 2.333, df = 2, *p* = 0.311) (Figure 2).

### 2.2. Metabolomic Profiling

**Differences in Metabolite Profiles of Leaves Between Varieties.** PCA is an unsupervised method which enables to identify the differences or similarities among samples. Charming Beauty and Robinetta differed in their leaf metabolomic profiles at all plant stages although the differences in flowers were relatively small (Figure 3A). The separation was mainly due to PC1 which explained 41% of the variation in leaf metabolites. The loading plot showed that the signals in the region between δ 1.92–0.80 ppm had a low score and thus were associated with Robinetta the WFT resistant variety (Figure 3B). The signals at δ 1.28 (signal A) and 0.90 ppm (signal B) were related to triterpenoid saponins. In this region we could further identify signals related to the amino acids valine (δ 1.06) alanine (δ 1.48), and threonine (δ 1.32). Signals with a high score on the loading plot, that thus were associated with Charming Beauty, were in the sugar region δ 5.0–3.0 ppm (Figure 3B). However, the relative concentrations of the sugars we could identify, sucrose (δ 5.40), α-glucose (δ 5.20) and β-glucose (δ 4.60) did not differ significantly between Charming Beauty and Robinetta (F = 1.284, df = 1, *p* = 0.272; F = 0.351, df = 1, *p* = 0.561 and F = 0.219, df = 1, *p* = 0.645, respectively) (Figure 4). The relative concentrations of the triterpenoid saponins that were related to signal A and signal B were significantly higher in Robinetta (H = 16.323, df = 1, *p* = 0.000 and H = 14.449, df = 1, *p* = 0.000, respectively) than in Charming Beauty (Figure 4). The relative concentrations of alanine, valine and threonine were about three to four times higher in Robinetta than in Charming Beauty (F = 73.702, df = 1, *p* = 0.000; F = 334.108, df = 1, *p* = 0.000; F = 584.607, df = 1, *p* = 0.000, respectively) (Figure 4).

**Differences in Metabolite Profiles between Plant Organs.** The PCA analysis of the metabolomic profiles of the three plant organs showed clear differences for Charming Beauty (Figure 5A). PC1, which explained 42% of the variation, separated the flowers from the leaves and buds. The loading plot for PC1 showed that the region between δ 5.40–3.00 ppm which represents sugar compounds was responsible for this separation (Figure 5B). In Robinetta too plant organs were separated by their metabolomics profiles in the PCA (Figure 6A). The separation was mainly due to PC1 which explained 57% of the variation in plant metabolites. Signals with low values on the loading plot, and thus associated with buds and leaves belonged to the region δ 2.5–0.80 ppm. These signals were related to amino acids and saponins. Other signals with a negative value on the loading plot in the region δ 4.20–3.20 ppm, which we identified as being from sucrose, were associated with buds and leaves. Signals with positive values on the loading plot and thus associated with flowers, in the range from δ 4.00–3.28 ppm (Figure 6B) were identified as glucose.

The relative concentration of signal A did not show significant differences among plant organs in Charming Beauty (H = 2.333, df = 2, *p* = 0.311). The relative concentration of signal B was slightly higher in buds compared to flowers and leaves (H = 6.706, df = 2, *p* = 0.035). Threonine, alanine and valine were two times higher in buds and flowers in Charming Beauty than in leaves (F = 5.335, df = 2, *p* = 0.039; F = 29.535, df = 2, *p* = 0.000; F = 16.347, df = 2, *p* = 0.002, respectively) (Figure 7). The concentrations of α- and β-glucose were about two times higher in flowers than in leaves and buds (F = 31.846, df = 2, *p* = 0.000 and F = 27.131, df = 2, *p* = 0.001, respectively) (Figure 7). However, the relative concentration of sucrose (δ 5.40 ppm) was lower in flowers than in leaves and buds (F = 5.502, df = 2, *p* = 0.020) (Figure 7).

In Robinetta signals A and signal B were about 50% higher in leaves and buds than in flowers (F = 63.507, df = 2, *p* = 0.000 and F = 14.969, df = 2, *p* = 0.005, respectively). Threonine was higher in leaves and buds than flowers (F = 61.767, df = 2, *p* = 0.000). Alanine was similar in concentration in all plant organs (F = 3.056, df = 2, *p* = 0.122). Valine concentration was about 50% higher in leaves and buds (F = 7.368, df = 2, *p* = 0.004) than in flowers (Figure 7). Relative concentrations of α- and β-glucose were about two times higher in flowers than in leaves and buds (F = 5.543, df = 2, *p* = 0.043 and F = 404.909, df = 2, *p* = 0.000, respectively) (Figure 7). In contrast, sucrose was lower in flowers than in leaves and buds (F = 10.648, df = 2, *p* = 0.011) (Figure 7).

**Differences in Metabolite Profiles** of **Leaves between Plant Development Stages.** The PCA analysis of metabolomic profiles did not separate the leaves of the three developmental stages in both Charming Beauty and Robinetta. In addition, the relative concentrations of the two triterpenoid saponins (signal A and signal B), the amino acids and the sugars did not differ among the leaves from different plant developmental stages.

**Differences in Metabolite Profiling between Environmental Conditions.** Visual inspection of the NMR-metabolomic profiles of plants grown under different environmental conditions (field, field transition and climate room) clearly showed differences between varieties and among growing conditions (Figure 8). To further analyze these results multivariate data analysis was applied. First principal component analysis (PCA) was used. However, there was no clear clustering of the different samples within each variety. Apparently, the variability of the samples was too high to give a clear separation. Using the three growing conditions we then applied PLS-DA, for each variety. The climate room grown samples clearly separated from the other two groups of field grown plants and plants transferred from the field to the climate room. The latter two overlapped in the PLS-DA scoring plots of both Charming Beauty (Figure 9A) and Robinetta (Figure 9B). The first component explained 80% and 81% of the variance in the dataset in Charming Beauty and in Robinetta, respectively. The climate chamber-grown plants were clustered at the negative side of PC1.

The important question one may ask is if there is a consistent difference between the two varieties independent of the environmental conditions. All compounds known to be associated with thrips resistance in *Gladiolus* [18] were higher in Robinetta, the resistant variety, for all three environmental conditions (Figure 10). Between environmental conditions there were some metabolomic differences, with a trend for triterpenoids to be lower under climate room conditions. A similar trend seemed to be present for the amino acids alanine, valine and threonine and sucrose (Figure 10). In contrast the concentrations of kaempferol were significantly higher when plants were grown in the climate room.

All compounds known to be associated with susceptibility to thrips [18] were higher in Charming Beauty, the susceptible variety, for all three environmental conditions (Figure 11). Concentrations of α-glucose and β-glucose were lower in the climate room whereas concentrations of gallic acid and epigallocatechin were higher while the concentration of epicatechin was not affected by environmental conditions.

Other metabolites that changed due to different environmental were luteolin and apigenin (Figure 12) as well as the organic acids formic- and malic acid (Figure 12). Concentrations of luteolin and apigenin were significantly higher in field grown plants while concentrations of formic- and malic acid were higher in the climate chamber.

## 3. Discussion

Robinetta and Charming Beauty showed consistent differences in WFT resistance over all development stages. Robinetta as the resistant variety exhibited more than 500-fold less silver damage at all plant development stages compared to Charming Beauty. Metabolomic profiles differed between the two varieties throughout all three plant stages. They revealed triterpenoid saponins and amino acids as metabolites associated with the resistant variety as shown in our earlier study [18]. Those compounds were consistently higher in Robinetta overall plant stages. Threonine was 10 times higher and triterpenoid saponins, valine and alanine were about five times higher in Robinetta. With the exception of valine all these compounds were observed to be negatively correlated with thrips resistance.

In Charming Beauty leaves were more damaged compared to buds and flowers: 50% of all damage occurred on the leaves. Metabolomic profiles differed among plant organs. Triterpenoid saponins were slightly higher in buds and amino acids were two to three times higher in buds and flowers compared to leaves. Patterns in metabolites related to resistance were, therefore, in line with patterns in silver damage. However, leaves represent a relatively larger area compared to buds and flowers so that differences in damage between organs may not solely be attributed to variation in metabolites. Although the silver damage on leaves was higher in the vegetative stage than in the two generative stages, we did not observe significant differences in leaf metabolites related to resistance (or to susceptibility) between leaves of different developmental stages. While in Robinetta damage was always much lower than in Charming Beauty, the concentrations of all compounds identified as being related to thrips resistance where much higher. In Robinetta, the relative concentrations of the triterpenoid saponins (signals A and B) of threonine, and of valine were much higher in leaves and buds than in flowers.

Whereas in many plants species old leaves are more attractive to WFT than young leaves we observed an opposite pattern in *Gladiolus* [13]. Damage to leaves was highest in the vegetative life-stage when leaves were on average young. However, vegetative and generative plant stages have similar leaf numbers while leaf area expands with age. Moreover, the concentration of defence compounds in leaves did not drop during successive life-stages. Having a higher concentration of defence compounds in buds and flowers is a way to protect the most valuable organs with respect to plant fitness from WFT. Similarly, Damle et al. [9] reported an accumulation of proteinase inhibitors in flowers as a protection against *Helicoverpa armigera* on tomato (*Lycopersicon esculentum* Mill). The pattern of damage across plant organs in Charming Beauty contrasted with the ornamental chrysanthemum, on which WFT preferred flowers over leaves. In the latter species WFT is attracted to pollen and it may find shelter in the flowers. In contrast to what we observed for Gladiolus, WFT caused more damage on plants with flower buds, than on plants with fully opened flowers or on plants with only leaves in *Impatiens walleriana* [10].

Differences in WFT resistance between the susceptible variety Charming Beauty and the resistant variety Robinetta remained constant across developmental stages. Furthermore, the concentrations of leaf metabolites identified to be associated with resistance in our earlier study [18] remained similar during the different development stages for both varieties. These results strongly suggest that markers for resistance in early developmental stages remain valid throughout the plant’s life.

The effect of the environment on the metabolomic profile is clear between plants grown in the field and in the climate room, but the transition from the field into the climate chamber does not seem to cause many changes in the metabolome. Metabolites that were affected by the growing conditions included the flavonoids kaempferol, apigenin, and luteolin, as well as some organic acids: formic acid, gallic acid and malic acid. Climate room generally have a lower photosynthetic active radiation (PAR) level and UV-B dose compared to field conditions [19]. In the present study, light in the climate chamber was lower than in field conditions which might have caused the chemical variation. Kaempferol was at higher levels in the climate room grown plants. This is in accordance with the results reported by Muller et al. [20] for the perennial semi-aquatic plant *Hydrocotyle leucocephala* showing higher kaempferol concentrations for plants grown in climate room compared to plants grown in natural light conditions in the field. In contrast, luteolin and apigenin, were higher in field and field transition-grown plants. Markham et al. [21], reported that in the thallus of the common liverwort, *Marchantia polymorpha* the flavonoids, luteoline and apigenin, had a strong positive correlation to UV-B levels. Formic acid, gallic acid and malic acid were higher in climate room-grown plants whereas Jankanpaa et al. [2] reported that malic acid was more abundant in high-light plants than in low-light plants of *Arabidopsis*.

Concentrations of metabolites previously found to be related to thrips resistance were similar in each of the three environments while differences between the two varieties remained. Consequently, the environment seemed no to have affected the compounds related to constitutive thrips resistance in *Gladiolus*. In other words, resistance in *Gladiolus* seems mainly genetically determined.

Unlike secondary metabolites, amino acids belong to the primary metabolites and are part of the plants primary metabolism which is responsible for plant growth and development. Amino acids were reported by Jankanpaa et al. [2] as light-intensity dependent compounds in *Arabidopsis thaliana*. Valine was strikingly higher in plants grown under low light (30 µmol photons m^−2^ s^−1^) conditions, alanine had higher concentrations in high light (600 µmol photons m^−2^ s^−1^) and normal light (300 µmol photons m^−2^ s^−1^). Threonine had accumulated in Arabidopsis one hour after transfer from a growth chamber into the field. In the present study, alanine, valine and threonine were slightly lower in the climate chamber with lower light intensity (Figure 11).

All together our results show that differences in plant defence compounds related to thrips resistance between a resistant and a susceptible variety persist during plant development and under different growing conditions. Therefore, they seem useful for breeding programs targeted at resistance. However, when breeding for resistance it is important not to impair bulb or flower production. These metabolites associated with resistance are among the most expensive defence metabolites (triterpenoid saponins) for plants to synthesize [22]. Thus, the higher expenditure in resistance may be one of the factors leading to a smaller dry mass of Robinetta compared to Charming Beauty [18]. More research on the costs of resistance would be needed for a successful breeding program.

## 4. Materials and Methods

### 4.1. Plant Materials

Two *Gladiolus nanus* varieties, (Charming Beauty and Robinetta), from vegetative, generative with buds and generative with flowers stages were obtained from the *Gladiolus* breeder Gebr. P. & M. Hermans (Lisse, The Netherlands).

### 4.2. Plant Development Stages

We grew plants outdoors in a field at Lisse, the Netherlands, to mimic the natural growing conditions. Plants at three development stages, i.e., vegetative, generative with buds and generative with flowers were collected from the field by carefully digging out the plants with their root system. Consequently, they were then potted and placed in a climate room (L:D, 18:6, 20 °C) for 7 days of further growth before they were infested by thrips. Robinetta was planted in the field 25 days earlier as Charming Beauty on May 2013. Because we harvested all the plants in a particular stage at the same day Robinetta plants had been in the field for a longer time period. Vegetative plants of Charming Beauty and Robinetta were thus collected after 65- and 90-days growth in the field, respectively. Plants with buds that just started to develop were collected after 75 days and 100 days in the field, respectively and plants with fully developed buds that started to open flowers were collected after 85 and 110 days, respectively. After collecting, plants were transferred to a climate chamber. Four to six replicates of all development stages were used for NMR metabolomics.

### 4.3. Different Environmental Conditions

Vegetative plants were grown under three different conditions: field, field transition and climate chamber. These plants were planted as bulbs to 9 × 9 cm pots filled with a 1:1 mixture of potting soil and dune sand. They were randomly placed in a climate chamber (L:D, 18:6, 20 °C, 70% relative humidity and 90–120 µmol photons m^−2^ s^−1^) and grown for 70 days. Field-grown plants were planted and grown for 65 days (Charming Beauty) and 90 days (Robinetta). Part of these were carefully dug out from the field and transferred immediately into a climate room for 7 days. Plants from all conditions were harvested at the vegetative stage. Four to six replicates of all three conditions were used for NMR metabolomics.

### 4.4. Thrips Whole Plant Bioassay

For each of the two varieties, three to four plants per developmental stage were tested in a non-choice whole plant bioassay. Each plant was placed individually in a WFT proof cage, consisting of a plastic cylinder (80 cm height, 20 cm diameter), closed with a displaceable ring of WFT proof gauze. The cages were arranged in a fully randomized design. Two adult males and 18 adult females of western flower WFT were released in each cage and left for 10 days. Thereafter, silver damage, expressed as the leaf area damaged in mm^2^, was visually scored for each plant. Silver damage in the buds and flowers in flowering plants were counted in mm^2^ [12].

We calculated total damage per plant as the sum of the silver damage in all plant organs present in a certain stage. Because WFT damage in Robinetta was zero in many samples we could not use a two-way ANOVA to test for the effects of variety and developmental stage on silver damage. Instead, we tested for the effects of developmental stage for each variety separately. We used the Kruskal-Wallis test to do so for Robinetta and we used one-way ANOVA for Charming Beauty. Differences in total damage between the two varieties were analyzed by using a Mann-Whitney U test.

### 4.5. Metabolomic Profiling

#### 4.5.1. Extraction of Plant Materials for NMR Metabolomics

The dried plant material was used to test for differences among leaves of the three developmental stages and for differences among buds and flowers in flowering plants for the two varieties using the standard protocol of sample preparation and ^1^HNMR profiling [23].

Samples of 30 mg freeze-dried plant material were weighed into a 2 mL microtube and extracted with 1.5 mL of a mixture of phosphate buffer (pH 6.0) in deuterium oxide containing 0.05% trimethylsilylproprionic acid sodium salt-*d*_4_ (TMSP) and methanol-*d*_4_ (1:1). Samples were vortexed at room temperature for 1 min, ultrasonicated for 20 min and centrifuged at 13,000× *g* rpm for 10 min. an aliquot of 0.8 mL of the supernatant were transferred to 5 mm NMR tubes for ^1^HNMR measurement.

#### 4.5.2. NMR Analysis

^1^HNMR spectra were recorded with a 500 MHz Bruker DMX-500 spectrometer (Bruker, Karlsruhe, Germany) operating at a proton NMR frequency of 500.13 MHz Deuterated methanol was used as the internal lock. Each ^1^HNMR spectrum consisted of 128 scans requiring 10 min and 26 s acquisition time with following parameters: 0.16 Hz/point, pulse width (PW) of 30 (11.3 µs), and relaxation delay (RD) of 1.5 s. A pre-saturation sequence was used to suppress the residual water signal with low power selective irradiation at the water frequency during the recycle delay. Free induction decay (FIDs) was Fourier transformed with a line broadening (LB) of 0.3 Hz. The resulting spectra were manually phased and baseline corrected to the internal standard TMSP at 0.00 ppm, using TOPSPIN (version 3.5, Bruker). Two-dimensional J-resolved NMR spectra were acquired using 8 scans per 128 increments for F1 and 8 k for F2 using spectral widths of 5000 Hz in F2 (chemical shift axis) and 66 Hz in F1 (spin-spin coupling constant axis). Both dimensions were multiplied by sine-bell functions (SSB = 0) prior to double complex Fourier transformation. J-resolved spectra were tilted by 45 o, symmetrized about F1, and then calibrated to TMSP, using XWIN NMR (version 3.5, Bruker). 1H-1H correlated COSY spectra were acquired with a 1.0 s relaxation delay and 6361 Hz spectral width in both dimensions. The window function for the COSY spectra was Qsine (SSB = 0).

#### 4.5.3. Data Processing

Spectral intensities were scaled to total intensity and reduced to integrated equal width (0.04 ppm) for the region of δ 0.32–10.0. The regions of δ 4.7–5.0 and δ 3.30–3.34 were excluded from analysis due to the presence of the residual signals of water and methanol. ^1^HNMR spectra were automatically binned by AMIX software (version 3.7, Biospin, Bruker). Plant development stages data were further analyzed with principal component analysis (PCA) performed with SIMCA-P software (version 15.0 Umetrics, Umea, Sweden). Pareto scaling was used for PCA analysis. With the PCA we tested for differences in metabolomics profiles between the two varieties. Besides, different environmental conditions data were further analyzed with partial least square-discriminant analysis (PLS-DA) which used unit variance scaling.

The peak area of triterpenoid saponins at δ 1.28 and 0.92 ppm earlier reported to be related to trips resistance in *Gladiolus* [18] were close to zero in all plant development stages in Charming Beauty. We, therefore, analyzed differences in these signals with the Kruskal-Wallis test, while all others were analyzed by one-way ANOVA. The relative concentrations of threonine, valine, alanine, sucrose, α-glucose and β-glucose were ln-transformed to fit a normal distribution. For leaves, differences between the two varieties in the peak areas of triterpenoid saponins were analyzed with a Kruskal-Wallis test, while differences between the two varieties in other metabolites were analyzed with one-way ANOVA. Differences in relative concentrations of triterpenoid saponins between plant organs were analyzed with a Kruskal-Wallis test while differences in other metabolites were analyzed with one-way ANOVA within variety. Differences in the relative concentrations of compounds between leaves at different developmental stages were analyzed with one-way ANOVA within variety. Data were subsequently analyzed with the Scheffe post-hoc test. Differences in metabolite concentrations between plants grown under different conditions, were analyzed separately using one-way ANOVA. Data was log-transformed to fit a normal distribution. Triterpenoid saponins, threonine and kaempferol were analyzed by Kruskal-Wallis tests.

## 5. Conclusions

In the present study metabolomic profiling is able to differ between the two varieties throughout all three plant stages. Differences in resistance between the susceptible variety Charming Beauty and the resistant variety Robinetta remained constant across developmental stages. Furthermore, we found no differences in resistance of leaves among developmental stages for both varieties which was accompanied by the absence of differences among developmental stages of metabolites in leaves that were identified as associated with resistance. Together, these results strongly suggest that markers for resistance in early developmental stages remain valid throughout the plant’s life.

## Figures and Tables

**Figure 1 plants-13-00687-f001:**
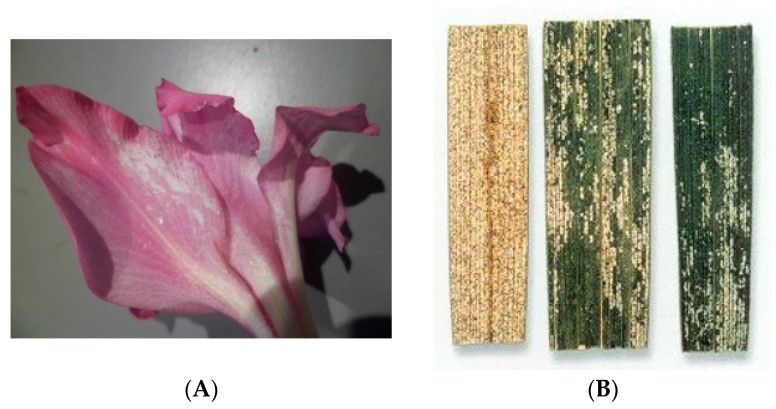
Plant silver damage in Charming Beauty on flowers (**A**) and leaves (**B**).

**Figure 2 plants-13-00687-f002:**
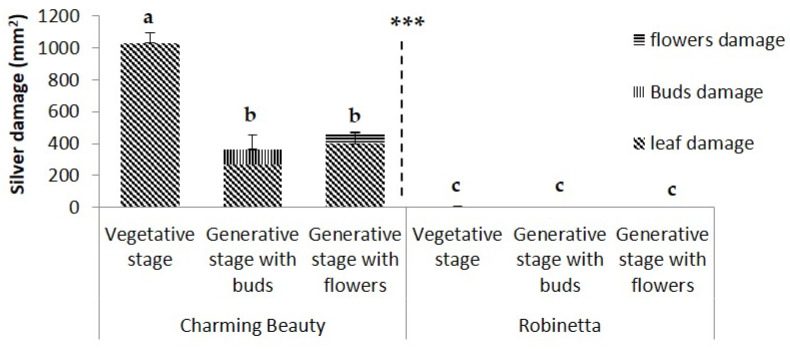
Plant silver damage (mm^2^) in Charming Beauty and Robinetta at three plant development stages: vegetative, generative with buds and generative with flowers as measured by a whole plant Western Flower Thrips non-choice bioassay. Bars represent total plant damage, patterns within bars represent different plant organs. Differences in total plant damage within the three developmental stages were tested with one-way ANOVA (Charming Beauty) and Kruskal-Wallis (Robinetta). Data represent mean and standard errors for three to four replicates. Different letters above the bars refer to significant differences within development stages at the 0.05 level. *** Indicate significant differences between the varieties (*p* < 0.000).

**Figure 3 plants-13-00687-f003:**
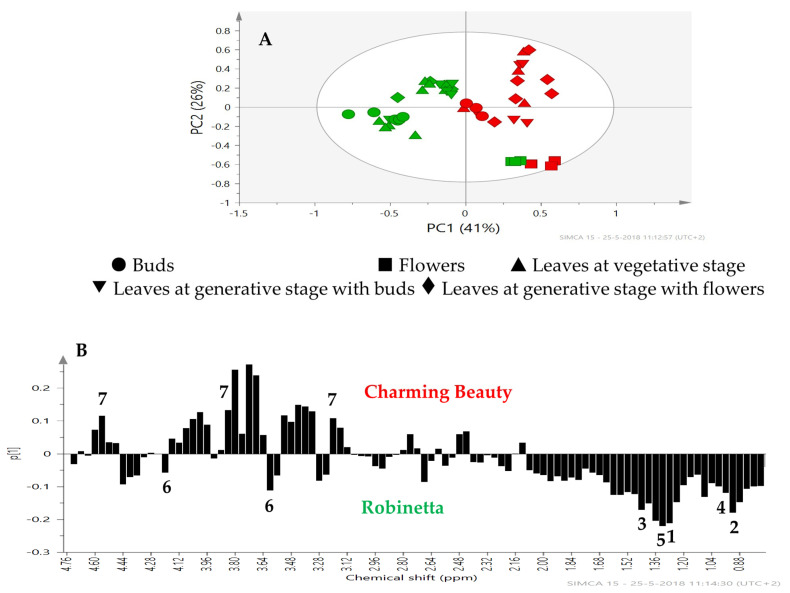
PCA score plot (**A**) and loading plot (**B**) for two varieties, Robinetta (green) and Charming Beauty (red) based on ^1^H NMR spectra with (●) buds, (■) flowers (▲) leaves at vegetative stage, (▼) leaves at generative stage with buds (♦) leaves at generative stage with flowers. Metabolites are labeled as triterpenoids saponins (1 and 2), alanine (3), valine (4), threonine (5), sucrose (6) and glucose (7).

**Figure 4 plants-13-00687-f004:**
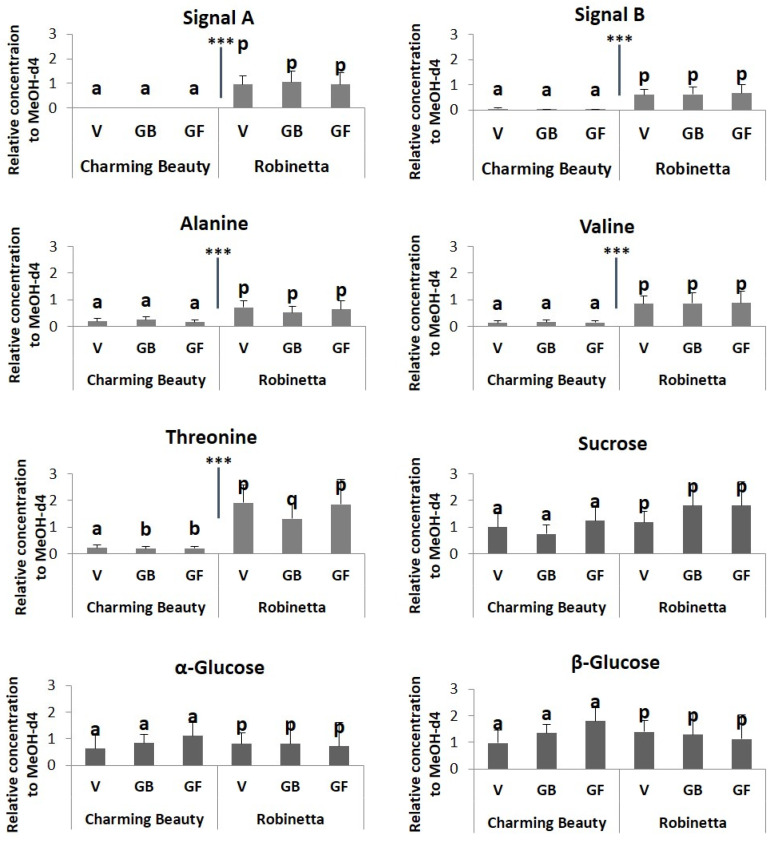
Relative concentration, as proportion of the internal standard, in ^1^H NMR spectra of triterpenoid saponins (signal A and signal B), alanine, valine, threonine, sucrose, α-glucose and β-glucose in leaves of three plant development stages (Vegetative (V), Generative with buds (GB), Generative with flowers (GF)) of Charming Beauty and Robinetta. Data present the mean ± SE of four to six for replicates of leaves at the vegetative, generative with buds and generative with flower stages. Differences in relative concentrations of triterpenoid saponins and amino acids within variety and between the two varieties were analyzed by a Kruskal-Wallis test and a one-way ANOVA, respectively. Differences in the relative concentrations of sucrose, α-glucose and β-glucose between the two varieties and within variety were analyzed by one-way ANOVA. Different letters refer to significant differences among development stages within varieties at the 0.05 level. *** indicate significant differences between varieties (*p* < 0.000).

**Figure 5 plants-13-00687-f005:**
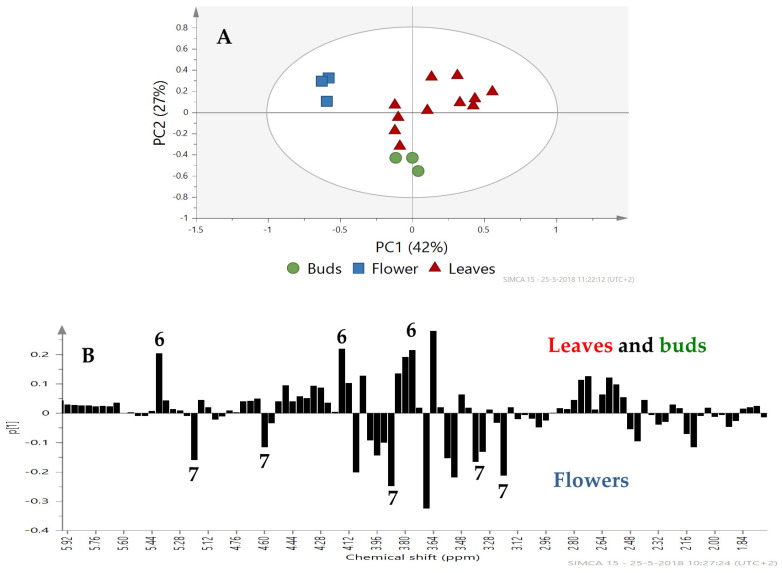
PCA score plot (**A**) and loading plot PC1 (**B**) for Charming Beauty based on ^1^H NMR spectra with (●) buds (■) flowers and (▲) leaves from plants at the generative stage. Metabolites are labeled as sucrose (6) and glucose (7).

**Figure 6 plants-13-00687-f006:**
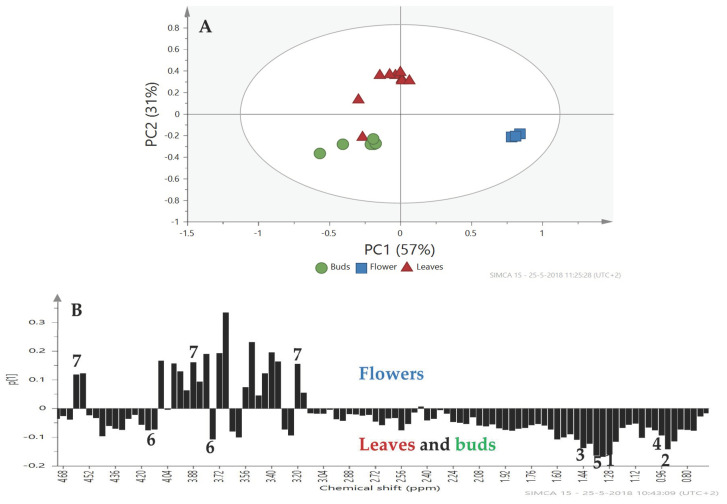
PCA score plot (**A**) and loading plot (**B**) for Robinetta based on ^1^H NMR spectra with (●) buds, (■) flowers and (▲) leaves from plants in the generative stage. Metabolites are labeled as signal A (1), signal B (2), alanine (3), valine (4), threonine (5), sucrose (6) and glucose (7).

**Figure 7 plants-13-00687-f007:**
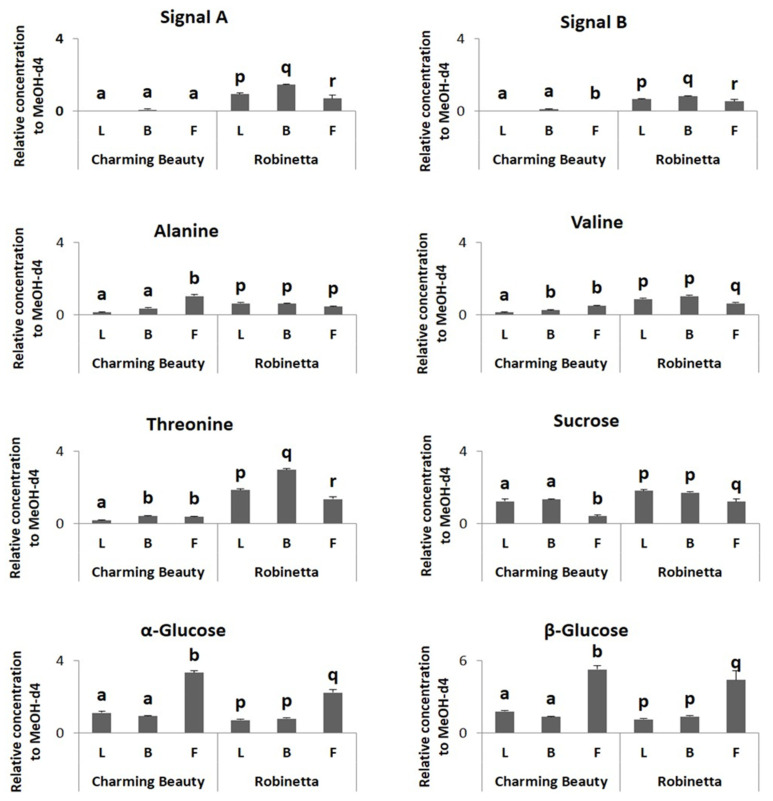
Relative concentrations, as proportions of the internal standard, in ^1^H NMR spectra of triterpenoids, threonine, valine, alanine, α-glucose and β-glucose in different plant organs (Leaves (L), Buds (B), Flowers (F)) of Charming Beauty and Robinetta. Data present the mean of four to six for replicates of plants in the generative stage ± SE of the mean. Differences in relative concentrations of triterpenoid saponins and amino acids within variety and between the two varieties were analyzed by Kruskal-Wallis test and one-way ANOVA, respectively. Different letters refer to significant differences among plant organs within varieties at the 0.05 level. Differences between varieties were not significant at the 0.05 level.

**Figure 8 plants-13-00687-f008:**
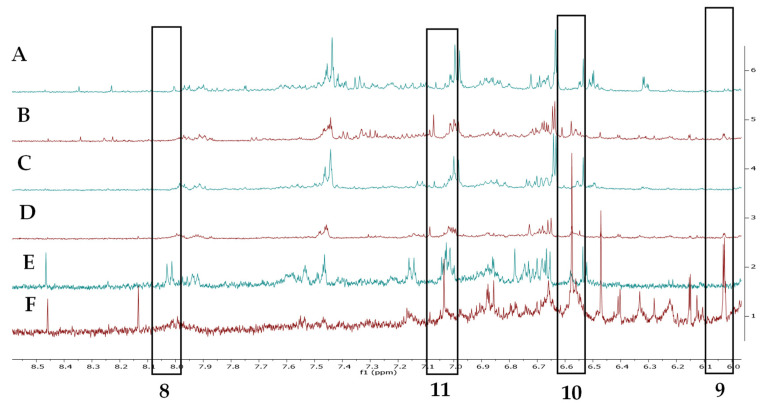
Comparison of ^1^H-NMR spectra in the phenolics regions of Robinetta (**A**) and Charming Beauty plants grown in the field, (**A**,**B**) and (**D**), grown in the field and transferred to a climate chamber (**C**,**D**) and grown in a climate chamber (**E**,**F**). Metabolites associated with resistance *Western Flower Thrips* are labelled as kaempferol (8), epicatechin (9), epigallocatechin (10) and gallic acid (11).

**Figure 9 plants-13-00687-f009:**
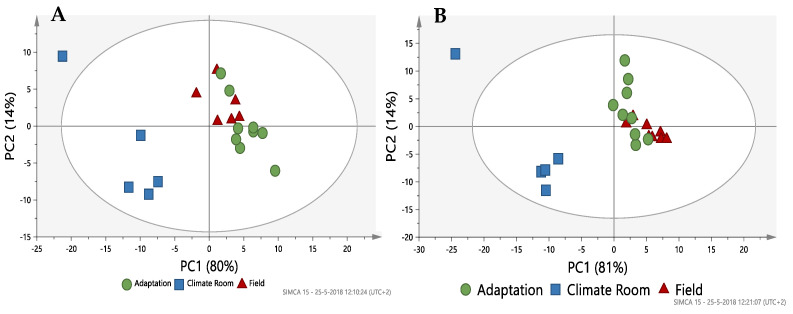
Score plot of PLS-DA based on Charming Beauty (**A**) and Robinetta (**B**) plants grown in the field (▲), plants grown in the field and transferred to a climate chamber (■) and plant grown in a climate room (●).

**Figure 10 plants-13-00687-f010:**
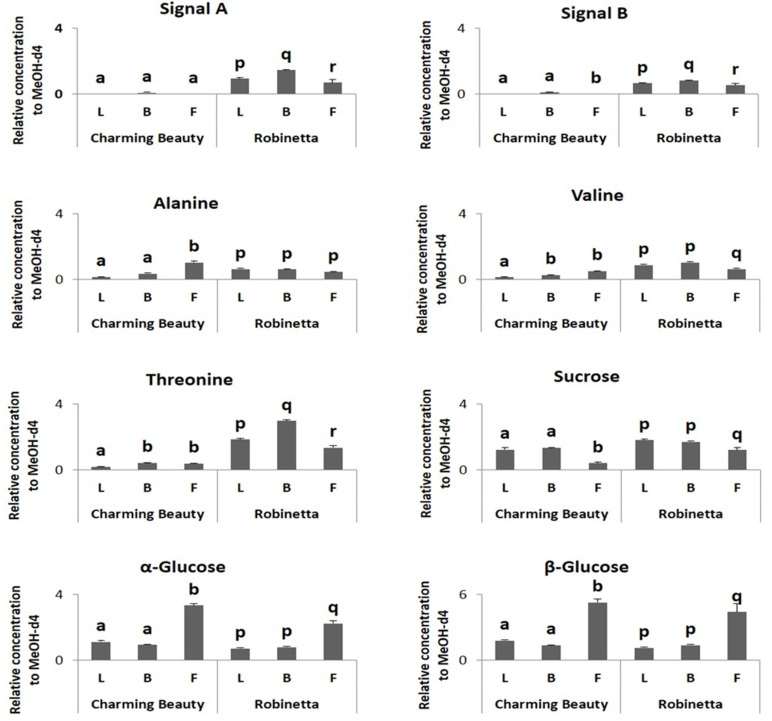
Relative concentrations, as proportions of the internal standard, in ^1^HNMR spectra of signal A, signal B, alanine, valine, threonine, sucrose and kaempferol related to different environmental conditions of the thrips susceptible variety Charming Beauty (CB) and the resistant variety Robinetta (R). Data present the mean of four to six replicates ± SE of the mean. Signal A, signal B, threonine and kaempferol were analyzed by Kruskal-Wallis test. Alanine, valine and sucrose were analyzed by one-way ANOVA. Different letters refer to significant differences between varieties in each environmental condition at the 0.05 level.

**Figure 11 plants-13-00687-f011:**
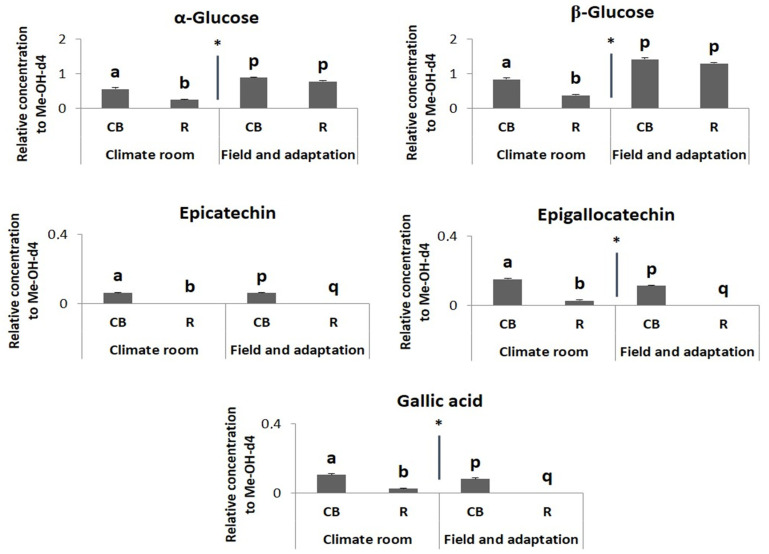
Relative concentrations, as proportions of the internal standard, in ^1^HNMR spectra of α-glucose, β-glucose, epicatechin, epigallocatechin and gallic acid as the metabolites associated with the kaempferol related to different environmental conditions of the thrips susceptible variety Charming Beauty (CB) and the resistant variety Robinetta (R). Data present the mean of four to six replicates ± SE of the mean. α-glucose and β-glucose were analyzed by one-way ANOVA while epicatechin, epigallocatechin, gallic acid and kaempferol were analyzed by Kruskal-Wallis test. Different letters refer to significant differences between varieties in each environmental condition at the 0.05 level, while * indicate significant differences between environmental conditions at the 0.05 level.

**Figure 12 plants-13-00687-f012:**
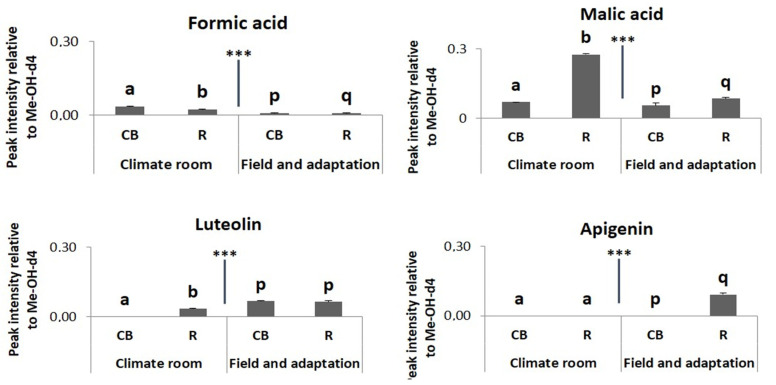
Relative concentrations, as proportions of the internal standard, in ^1^HNMR spectra of formic acid, malic acid, luteolin and apigenin kaempferol related to different environmental conditions of the thrips susceptible variety Charming Beauty (CB) and the resistant variety Robinetta (R). Data present the mean of four to six replicates ± SE of the mean. Formic acid and malic acid were analyzed by one-way ANOVA while luteolin and apigenin were analyzed by Kruskal-Wallis tests. Different letters refer to significant differences between varieties in each environmental condition at the 0.05 level. *** Indicate significant differences between environmental conditions (*p* < 0.000).

## Data Availability

^1^H NMR data of the samples are deposited in the data storage of Natural Products Laboratory (Institute of Biology, Leiden University, Leiden, The Netherlands). The data can be provided upon request.

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
