# Peer review of "Resistance to Frankliniella occidentalis during Different Plant Life Stages and under Different Environmental Conditions in the Ornamental Gladiolus"

_plants, 2024, doi:10.3390/plants13050687_

Round 1
Reviewer 1 Report
Comments and Suggestions for Authors
First of all, I would like to congratulate the authors on the manuscript submitted.
I only have a few questions regarding to the experiments carried out, in particular, the NMR analysis for metabolite profiling and the multivariate methods to analyzing the experimental data obtained.
1) The authors has cited their previous study where are presented some metabolites related to resistance and susceptibility of Gladiolus varieties to western flower thrips using a NMR-based metabolite profiling approach, mainly triterpenoid saponins and amino acids. However, in the Results and Discusion sections they state regarding diferences on phenolic compounds, in particular flavonoids, such as kaempferol, apigenin and lutolin. My questions is: How were flavonoids identified by the 1H-NMR signals present in the analyzed extracts?
2) In Lines 269 and 270, where the authors present their results regarding differences in metabolite profiling between environmental conditions, it is stated: "First principal component analysis (PCA), was used. However, there was no clear clustering of the different samples within each variety". Considering this sentence, I ask the the authors: Why did you not use an unsupervised multivariate clustering method, in particular Hierarchical Cluster Analysis (HCA), since PCA is intended for exploratory analysis?
In addition to these questions, my other comments are related to small punctuation errors that I observed throughout the text and about the information presented in the title of Figure 9, which differs from what is presented in the text and in the graph caption. Ponctuation erros: Line 101 (there is an open paranthesis); Line 105 (there is a comma missing between vegetative and generative).
Author Response
Dear Reviewer,
Please see the attachment.
Kind regards,
Dinar

Reviewer 2 Report
Comments and Suggestions for Authors
The manuscript written by Wahyuni et al. is a descriptive work to gain a better understanding of the metabolic response in different tissues of two Gladiolus variants grown in the laboratory and in the field after attachment of western flower thrips. The research question is topical and has merit. However, it is very difficult to follow the text. There are too many pictures and the picture quality needs to be improved. So I suggest rewriting the text and shortening the number of illustrations.
Small comments:
Line 69, Line 99, Line 439, Line 521: The full stop is missing from the sentence.
Line 110-122: A rather unusual way of introducing the questions the study aims to answer.
Section 2.1: My suggestion is to include some photos of the plants, at least as a supplement, to show the observed differences in damage.
Section 2.1: Can you describe in the text the proportion between damaged and undamaged areas?
Section 2.2: Please briefly describe how the metabolomics study was performed in the text at the beginning of the section. I suggest showing representative NMR spectra of the varieties and tissues.
There is two Figure 4. It is a little bit confusing.
Line 168-172: The corresponding figure has not been referred to the text.
Line 265: There is a space in the text.
For example, in Figure 4 it is written vegetative and in Figure 6 it is written leaves. This is also true for generative with buds vs buds and generative with flowers vs flowers and flowers. Please describe the differences in the text.
Section “Differences in MetaboliteProfiling between Environmental Conditions”: Have the plants been treated with thrips? If so, please mention this in the text for better understanding.
Author Response
Dear Reviewer,,
Please see the attachment.
Kind regards
